# Applications of the Methylotrophic Yeast *Komagataella phaffii* in the Context of Modern Biotechnology

**DOI:** 10.3390/jof10060411

**Published:** 2024-06-06

**Authors:** Lidia Maria Pepe de Moraes, Henrique Fetzner Marques, Viviane Castelo Branco Reis, Cintia Marques Coelho, Matheus de Castro Leitão, Alexsandro Sobreira Galdino, Thais Paiva Porto de Souza, Luiza Cesca Piva, Ana Laura Alfonso Perez, Débora Trichez, João Ricardo Moreira de Almeida, Janice Lisboa De Marco, Fernando Araripe Gonçalves Torres

**Affiliations:** 1Laboratory of Molecular Biology, Department of Cell Biology, Institute of Biological Sciences, University of Brasília, Brasília 70910-900, DF, Brazil; lmoraes@unb.br (L.M.P.d.M.); henrique.marques@aluno.unb.br (H.F.M.); piva.luiza@gmail.com (L.C.P.); ana.alfonso1991@gmail.com (A.L.A.P.); janicedemarco@unb.br (J.L.D.M.); 2Laboratory of Genetics and Biotechnology, Embresa Brasileira de Pesquisa Agropecuária (EMBRAPA) Agroenergy, Brasília 70770-901, DF, Brazil; vivianereis.phd@gmail.com (V.C.B.R.); debora_trichez@yahoo.com.br (D.T.); joao.almeida@embrapa.br (J.R.M.d.A.); 3Laboratory of Synthetic Biology, Department of Genetics and Morphology, Institute of Biological Sciences, University of Brasília, Brasília 70910-900, DF, Brazil; cintiacoelhom@unb.br (C.M.C.); leitao.math@gmail.com (M.d.C.L.); 4Microbial Biotechnology Laboratory, Federal University of São João Del-Rei, Divinópolis 35501-296, MG, Brazil; asgaldino@ufsj.edu.br (A.S.G.); thaispaiva@ufsj.edu.br (T.P.P.d.S.)

**Keywords:** *Komagataella phaffii*, synthetic biology, renewable chemicals, biomaterials, biopharmaceuticals, biomaterials

## Abstract

*Komagataella phaffii* (formerly *Pichia pastoris*) is a methylotrophic yeast widely used in laboratories around the world to produce recombinant proteins. Given its advantageous features, it has also gained much interest in the context of modern biotechnology. In this review, we present the utilization of *K. phaffii* as a platform to produce several products of economic interest such as biopharmaceuticals, renewable chemicals, fuels, biomaterials, and food/feed products. Finally, we present synthetic biology approaches currently used for strain engineering, aiming at the production of new bioproducts.

## 1. Introduction

In the context of industrial biotechnology, many bio-based products can be obtained as a result of bioprocesses using microorganisms as cell factories. Among these, yeasts stand out as one of the most important microbial platforms. Despite the fact that baker’s yeast *Saccharomyces cerevisiae* still occupies a relevant position as one of the main cell factories [1], the methylotrophic yeast *Komagataella phaffii* has gained much attention as a promising “biotech yeast” [2].

*K. phaffii* exhibits traits ideal for a microbial platform in biotechnological settings. This yeast shows minimal nutritional needs and grows on economical substrates, achieving cell densities exceeding 100 g·L^−1^ of dry cell weight [3]. Notably, it demonstrates resilience against high methanol concentrations, acidic and basic pH levels, and inhibitors derived from lignocellulosic sources, surpassing other methylotrophic yeasts [4]. Moreover, it outperforms *S. cerevisiae* in terms of thermo- and osmo-tolerance [2]. Its appeal as a host organism is further accentuated by efficient secretion mechanisms, yielding abundant protein secretion in bioreactor settings [5], and versatility in protein processing and post-translational modifications [6], due to its ability to perform modifications such as *O-* and *N*-linked glycosylation and disulfide bond formation [7]. Also, *K. phaffii* can generate recombinant proteins either constitutively or through induction [2], facilitated by the development of numerous genetic and metabolic engineering tools alongside established fermentation processes [8].

Given its importance in modern biotechnology, in this review we present the main applications of *K. phaffii* in different industrial sectors. Finally, we show how modern synthetic biology approaches may be used to further optimize the utilization of *K. phaffii* in the context of modern biotechnology.

## 2. Applications in the Pharmaceutical Industry

The expression system based on *K. phaffii* is licensed in more than 100 companies involving the biotech, pharmaceutical, vaccine, and food industries [9]. Proteins produced in this yeast have also received GRAS (generally recognized as safe) status by the American Food and Drug Administration since 2006 [10] and *K. phaffii* was first approved for the production of biopharmaceuticals in the USA in 2009 [11]. Since then, it has become a promising platform to produce biopharmaceuticals, which include a wide range of products such as vaccines, blood and blood components, and recombinant therapeutic proteins [12]. A list of the main biopharmaceuticals produced in *K. phaffii* is shown in Table 1.

Ecallantide (trade name, Kalbitor) was the first FDA-approved biopharmaceutical produced in *K. phaffii*. It is a kallikrein inhibitor indicated for treatment of hereditary angioedema. In its discovery, researchers used a phage display technique to identify a possible human inhibitor analog that could interfere with inflammatory and coagulation pathways and expressed the recombinant 60-amino-acid protein in *K. phaffii* [32]. Dyax Inc., its original developer and manufacturer, has since been acquired by Shire, which in turn was acquired by Takeda Pharmaceuticals, currently the supplier of this biopharmaceutical to the USA [13]. Kalbitor raised some concerns regarding hypersensitivity reactions, which hindered its approval by the European Medicines Agency [33].

Ocriplasmin (trade name, Jetrea), produced in *K. phaffii*, was approved by the FDA for the treatment of vitreomacular adhesion in 2012 [15]. This protein is a truncated form of human plasmin, a serine protease that acts on fibronectin and laminin. This proteolytic activity was shown to be able to resolve vitreomacular traction and reduce the requirement for surgical treatment [34]. Jetrea was initially approved for manufacturing by Thrombogenics but was discontinued in the USA due to commercial reasons; its rights are currently licensed to Inceptua Pharma in Europe [14].

In 2021, the FDA approved the first biosimilar insulin product in the American market, insulin glargine-yfgn, also produced in *K. phaffii* (Semglee, Mylan Pharmaceuticals Inc., now a part of Viatris Inc.) [16]. Insulin glargine-yfgn is an insulin analogue with a prolonged duration of action that is also approved by the European Medicines Agency [17]. This protein showed noninferiority versus the reference insulin glargine and, as a biosimilar, had a reduced price and significantly improved access to diabetes treatment [35]. Biosimilar insulin products produced in this yeast and approved for commercialization in Europe include Baxter’s Inpremzia and Mylan’s Kirsty, which have shown comparable results to Novo Nordisk’s reference biopharmaceuticals, Actrapid and Novorapid [20,21], respectively.

The production of biopharmaceuticals in *K. phaffii* has bloomed in India and Japan owing to less strict intellectual property regulations than in the USA or Europe [28], and this market currently presents a range of *K. phaffii*-based products that include collagen, interferon, vaccines, and hormones [36,37]. Up until 2022, the Japanese Mitsubishi Pharma Corporation commercialized recombinant human serum albumin produced through a *K. phaffii*-based expression platform [38]. Human serum proteins comprise a large share of biopharmaceuticals expressed in *K. phaffii*: the “a” subunit of coagulation factor XIII, involved in coagulation disorders, and human antithrombin III, used in the treatment of disseminated intravascular coagulation, have been produced and purified in this yeast [39,40].

Research on the production of some biopharmaceuticals in *K. phaffii*, such as glycoproteins and antibodies, has faced issues regarding the fungal pattern of glycosylation of the resulting proteins, which could hinder their biological activity. Efforts towards engineering the glycosylation pattern of this yeast have resulted in Merck’s Glycofi strain, which produces proteins with human-like *N*-glycosylation and terminal sialylation. With this, the yeast could potentially replace mammalian cell lines in the production of human glycoproteins [41,42]. Different works describe the use of this platform to produce antibodies and erythropoietin [43,44,45]. Also, an anti-HER2 cancer therapeutic antibody produced in a Glycofi strain showed promising results in a preclinical study [46]. Unfortunately, the Glycofi facility was closed when Merck decided to stand back from biopharmaceutical research and development [47] in 2016.

Many other proteins produced in *K. phaffii* are currently under research, including pre-clinical and clinical trials. Fusions of human serum albumin and peptide hormones such as the parathyroid hormone (PTH) aim to increase the stability of these hormones and improve their pharmacokinetic properties via the increased biological half-life of albumin [48]. Considering other protein fusion technologies, antibody-directed enzyme prodrug therapy (ADEPT) represents an ingenious strategy for overcoming both drug resistance and lack of selectivity in anti-cancer treatments. An antibody directed against cancer cell antigens is fused with a drug-converting enzyme; the corresponding prodrug is then administered to the patient and converted into an active drug in the tumor, avoiding systemic toxicity. *K. phaffii* is used to produce a fusion currently undergoing clinical trials, MFECP1, which contains an anti-carcinoembryonic antigen antibody fused to a carboxipeptidase. Once the antibody tags cancer cells, a bis-iodo phenol mustard prodrug is administered and then converted by the peptidase, leading to cell death [25].

Since the FDA approval for the first therapeutic antibody in 1986, these proteins have set remarkable milestones in the treatment of various diseases, including cancer, immune disorders, and infectious diseases [49]. *K. phaffii* has been used to produce a few of them, some that are currently approved for commercialization and others still under research. Eptinezumab (trade name Vyepti), produced by Lundbeck Seattle BioPharmaceuticals, received approval by the American regulatory agency in 2020 and by the European agency in 2022 for the prevention of migraine. The rationale behind this is based on the fact that migraine is a neurovascular disorder involving the release of the vasodilator calcitonin gene-related peptide (CGRP). The *K. phaffii*-produced eptinezumab is a humanized monoclonal antibody that binds to CGRP, preventing the triggering of migraine episodes [50]. Clazakizumab is an anti-IL-6 antibody currently being studied in various clinical trials [27]. The ubiquitous role of IL-6 in immune disorders, inflammatory diseases, and even cancer has prompted the development of various studies with anti-IL-6 antibodies. Bristol-Myers Squibb acquired exclusive worldwide development rights for most applications of the biopharmaceutical initially developed in *K. phaffii* by Alder Pharmaceuticals and is currently conducting a phase II trial [51,52]. New applications of therapeutic antibodies include nanobodies, which are heavy chain domains of camelid antibodies that penetrate tissues and overcome the blood–brain barrier more efficiently than regular therapeutic antibodies owing to their small size. These proteins have been efficiently expressed in *K. phaffii*, and future studies on their glycosylation and binding properties should bring more insights to this subject [53,54]. Ablynx, now a part of Sanofi, holds the worldwide rights for the Nanobody trademark and carries out the research that could achieve future therapeutics using this protein platform [55].

Virus-like particles (VLPs) purified from *S. cerevisiae* are already FDA-approved and commercially available (for example, Gardasil and Gardasil9 against the human papilloma virus, HPV [56]). VLPs purified from *K. phaffii* are not yet available, but some examples undergoing preclinical studies include HPV and coxsackievirus [57]. DENV envelope protein-based VLPs generated using *K. phaffii* showed encouraging results against dengue [58,59,60] and gave the perspective of an inexpensive vaccine that could be used in developing countries where dengue is endemic. A preclinical evaluation of hepatitis B virus (HBV) core antigen VLPs purified from *K. phaffii* against hepatocellular carcinoma is also underway [61].

Specifically considering HBV, the recombinant hepatitis B surface antigen (rHBsAg) is expressed by *K. phaffii* in the production of Shanvac-B, a historically successful Indian vaccine indicated for immunization against chronic liver infection caused by all known subtypes of HBV. The antigen is produced by a culture of genetically engineered *K. phaffii* carrying the gene that codes for the major HBV surface antigen in a high-cell-density fed-batch fermentation process [62,63].

Other examples of recombinant vaccines with registered clinical trials (ClinicalTrials.gov, accessed 10 February 2024) include a hookworm vaccine tested in Brazilian and American participants [29], an intestinal schistosomiasis vaccine [29], and a recombinant malaria vaccine [31].

In view of the wide applicability of *K. phaffii* in the production of recombinant antigens and the increasing emergence of biological therapeutics against infectious, autoimmune, and non-communicable diseases, this production platform will certainly play a key role in biopharmaceutical production in the near future.

## 3. Applications in the Production of Renewable Chemicals and Fuels

Bio-based chemicals, materials, and fuels produced from renewable biomass such as lignocellulose or even carbon dioxide (CO_2_) are becoming an interesting alternative to replace, at least in part, those derived from fossil feedstock through more sustainable processes [64]. *K. phaffii* can utilize different substrates including glucose, fructose, ethanol, methanol, glycerol, sorbitol, succinic acid, and acetic acid [65,66,67]. Recently, the ability of *K. phaffii* to utilize xylose via the oxidoreductive pathway at a slow rate was demonstrated [68]. Furthermore, genes involved in xylose metabolism have also been introduced in the yeast to improve xylose utilization, leading to higher assimilation of this hemicellulosic sugar [69,70]. Exploring their natural methanol utilization pathway (MUT), CO_2_ may also be used as a substrate. For instance, two approaches have been proposed: methanol can be synthesized via CO_2_ hydrogenation and be further metabolized by the MUT pathway [71] or alternatively, in a more elaborate strategy, CO_2_ can be directly assimilated by an autotroph-engineered yeast [72].

Among the chemical compounds produced by *K. phaffii* are organic acids [73], sugar alcohols [74], polyketides [75], terpenoids [76], biopolymers [77], and biofuels [78]. However, in general, they represent proof of concept since parameters such as titers, yields, and productivities are not suitable yet for application in large-scale industrial processes [79]. A summary of the main bioproducts is shown in Table 2.

One of the first metabolites produced in *K. phaffii* was S-adenosyl-L-methionine (SAM), a potential agent for human therapy that acts as a methyl donor and precursor of some amino acids and peptides such as cysteine and glutathione [93]. A recombinant yeast expressing the SAM2 synthase gene was constructed and further improved by the knock-out of the cystathionine synthase (CBS). As a result, 13.5 g·L^−1^ of SAM was obtained in a 5 L-fermenter using L-methionine as the substrate and methanol induction.

*K. phaffii* has been used to produce chemical building blocks like organic acids with a broad range of applications in industry (Table 2). For instance, to produce lactic acid, the lactate dehydrogenase gene (*LDH*) from *Bos taurus* was introduced in the yeast [73]. The additional expression of an endogenous lactate transporter and optimization of oxygenation conditions during cultivation on glycerol increased the yield significantly, rising from 10% up to 70% of the maximum theoretical yield [73]. Similarly, lactic acid production has also been reported for a recombinant *K. phaffii* strain carrying a multicopy integration of the *LDH* gene derived from *Leuconostoc mesenteroides*, albeit using methanol as a carbon source [80].

In another study, glucose and methanol were used as substrates to produce C4-dicarboxylic acids. The combined overexpression of pyruvate carboxylase (*PYC1)* and malate dehydrogenase (*MDH1)* genes led to the production of 0.76, 42.28, and 9.42 g·L^−1^ fumaric acid, malic acid and succinic acid, respectively, by the engineered yeast [81]. By integration of the malonyl-CoA reductase *MCR* gene from *Chloroflexus aurantiacus* into the *K. phaffii* genome, recently, the production of 3-hydroxypropionic acid (3-HP) has also been demonstrated [82]. Protein and metabolic engineering strategies were used to increase titer and productivity, resulting in strains capable of producing 24.75 to 37.05 g·L^−1^ of 3-HP on glycerol [82]. To explore xylose utilization by *K. phaffii*, the yeast was engineered to express different xylose dehydrogenase genes from bacteria and filamentous fungi. The best strain produced 37 and 11 g·L^−1^ of xylonic acid from xylose and sugarcane bagasse hydrolysate, respectively, in batch cultivation [70] (Table 2).

Synthesis of sugar alcohols such as xylitol and inositol were reported by Louie et al. [74] (Table 2). The biotransformation of xylose into xylitol was evaluated in recombinant *K. phaffii* harboring heterologous xylose reductase genes and the glucose dehydrogenase, *gdh*, from *Bacillus subtilis*. The highest conversion rates, with cells expressing the xylose reductase from *Scheffersomyces stipitis*, reached up to 80% and 70% with productivity values of 2.44 and 0.46 g·L^−1^·h^−1^ from xylose and a non-detoxified hemicellulose hydrolysate, respectively. The authors also demonstrated that the biocatalyst cells could be recycled in multiple rounds of biotransformation without significant loss of activity [74].

The ability of *K. phaffii* to produce bulk chemicals and biofuels has been proven in recent years (Table 2). A synthetic route for 2,3-butanediol (2,3-BD) production was implemented in the yeast through the overexpression of two heterologous enzymes: the α-acetolactate synthase AlsS and α-acetolactate decarboxylase AlsD from *B. subtilis*, with the final reaction catalyzed by an endogenous 2,3-BD dehydrogenase. A titer of 74.5 g·L^−1^ 2,3-BD was achieved in fed-batch cultivation using an optimized medium and glucose as substrate [85]. Isobutanol and isobutyl acetate are other examples of chemicals produced in *K. phaffii*, exploring the native L-valine biosynthetic pathway [78].

Fatty acids and derivatives are important raw materials to produce advanced oil-based chemicals. In the case of *K. phaffii*, it was shown that deleting two native fatty acyl-CoA synthetase genes improved free fatty acid (FFA) accumulation in the cells [86]. Later, to achieve a higher production of FFA from methanol, a global rewiring of the central metabolism was proposed to drive the carbon flux to the final product. As a result, 23.4 g·L^−1^ FFA was produced by the engineered strain during bioreactor cultivation [86]. Using this FFA-overproducing background, 35.2 and 90.8 mg·L^−1^ fatty alcohols were also obtained from methanol or glucose, respectively, after the expression of three additional heterologous enzymes—a carboxylic acid reductase, a 4′-phosphopantetheinyl transferase, and an alcohol dehydrogenase (Table 2) [86].

Other bio-products produced by engineered *K. phaffii* include terpenoids, polyketides, and biopolymers (Table 2). Considering the efficient isoprenoid metabolism and functional expression of proteins such as cytochrome P450 enzymes of *K. phaffii* [87], several studies have described the production of different terpenoids in yeast, for instance: dammarenediol-II [87], lycopene, β-carotene [76,88] and (+)-nootkatone [89]. Terpenoids are natural products with broad-range applications in pharmaceutical and industrial sectors due to their distinct biological activities and high bioavailability, being used, for example, as flavoring additives, antioxidants, antiaging agents, drugs, and antitumoral agents [94]. Polyketides are a class of secondary metabolites with bioactive properties that have relevant applications in the pharmaceutical industry [95]. Engineered overproducing *K. phaffii* for the production of 6-methyl salicylic acid [90], lovastatin, and monacolin J [75] are examples of polyketides already obtained using this host cell (Table 2).

Furthermore, the ability of *K. phaffii* to produce different biopolymers has been investigated (Table 2). Polyhydroxyalkanoates (PHAs) were accumulated up to 1% DCW in a recombinant *K. phaffii* expressing a heterologous PHA synthase targeted to the peroxisome [91]. The production of hyaluronic acid, a glycosaminoglycan used in pharmaceutical and medical formulations, was also achieved by overexpression of endogenous genes involved with hyaluronic acid synthesis, combined with heterologous expression of hyaluronan synthase and UDP-glucose dehydrogenase from *Xenopus laevis* [77]. Also recently, the production of chondroitin sulfate and heparin has been reported. The engineered and optimized strains produced around 2 g·L^−1^ of these compounds from methanol in fed-batch cultivation [65].

## 4. Applications in the Production of Biomaterials

A biomaterial is “any substance or substance of substance, other than drugs, of synthetic or natural origin, that can be used for any period of time, that augments or replaces partially or totally any tissue, organ or function of the body, to maintain or improve the individual’s quality of life’’ [96]. Indeed, investigations delve into harnessing this system for biomaterial production, encompassing extracellular polysaccharides and recombinant proteins for diagnostics, therapeutics, and potentially clinical tissue engineering [97]. Consequently, the characterization and utilization of biomaterials necessitate substantial quantities, posing a challenge for their cost-effective industrial-scale production. Hence, *K. phaffii* emerges as a proficient host for manufacturing a diverse array of biomaterials [98].

In this context, silks produced by arthropods, such as spiders, silkworms, dragonflies, and bees, among others, attract attention due to their mechanical and biocompatibility characteristics. Silk produced by silkworms has been widely used in the textile industry and as suture material for many years [99], especially due to its combination of remarkable strength and flexibility. In this sense, these biomaterials can be used to create threads, films, microcapsules, foams, sponges, hydrogels, and implantable materials for application in regenerative medicine, implant coating, and drug delivery [100,101].

Egg glue proteins (EGPs) produced by some insects can form a sticky substance that helps to adhere to surfaces in order to reduce the exposure of eggs to external factors such as wind or rain. In this context, the structure of silkworm EGP was elucidated, providing relevant information about its structure function for uses as a biomaterial [102]. The adhesive properties of the natural and recombinant protein were tested by expressing the EGPs in *K. phaffii* and *E. coli*. In view of this, the importance of the protein produced with glycosylations was shown, as the natural EGPs and those expressed in *K. phaffii* presented better adhesive properties when compared to non-glycosylated EGPs produced in *E. coli*. Indeed, many bioadhesive proteins are known to be glycosylated [103,104], and this often contributes to many properties of protein function such as folding, solubility, thermostability, and protection against proteolysis [105]. Therefore, the production of bioadhesive proteins becomes advantageous in the *K. phaffii* expression system.

Recombinant protein cp19k-MaSp1, combining cp19k from the adhesion complex of the barnacle species *Megabalanus rosa* and MaSp1 from *Nephila clavipes* dragline silk, was engineered and expressed in *K. phaffii*, yielding a protein content of 53.38 mg·L^−1^. This recombinant protein exhibited remarkable adhesion capabilities, surpassing individual proteins, demonstrating enhanced biocompatibility, mechanical resilience, and self-healing properties conducive to cell adhesion, proliferation, and growth, particularly for human umbilical vein endothelial cells (HUVECs) [106]. Additionally, another protein composite with bioadhesive traits was synthesized in *K. phaffii* [107]. This genetic engineering endeavor involved mussel foot proteins 3 and 5 (Mfp3, Mfp5) from *Mytilus californianus*, gas vesicle protein A (GvpA) from *Dolichospermum flosaquae* (a cyanobacterium), and the CsgA curli protein from *E. coli*. The synergistic properties of this chimera, coupled with post-translational modifications during yeast expression, resulted in robust protein adhesion, positioning it as a promising biomaterial for forthcoming biomedical applications.

From another perspective, collagen is a natural biopolymer of the extracellular matrix that makes up many structures in the body, namely the skin, muscles, bones, and cartilage. It is widely used in many fields such as biomedical applications and pharmaceutical and cosmetic industries [97]. In the biomedical field, its use includes wound dressings, suture material, tissues, and drug delivery, among others [108]. Like collagen, gelatin, produced from denatured collagen, also finds biomedical applications. Its preparation takes place from the hot acid or alkaline extraction of animal tissues [100]. However, the production of these biopolymers has some bottlenecks, as they come from animal tissues and, consequently, can cause possible allergic reactions, in addition to the transmission of pathogens [109]. Therefore, the production of recombinant human collagen has been explored over the years, in order to obtain a product with a higher yield and better stability and biocompatibility [110,111].

In addition, there is a complexity in the production of helical collagen, since its thermal stability depends on the hydroxylation of proline, present in its central helical molecule, which is only hydroxylated by the enzyme prolyl-4-hydroxylase (P4H), only present in systems of mammalian cell expression [112]. To overcome this problem, some groups also cloned and co-expressed the P4H enzyme in *K. phaffii*, but with low protein production [113].

Recombinant human-like collagen (RHLC) was expressed in *K. phaffii* [108], showing an expression titer for the extracellular medium of 2.33 g·L^−1^ and purification of 98% in 48 h, showing more efficiency than the extraction of animal tissues. Furthermore, RHLC showed stability at high temperatures and good biocompatibility, with potential application for industrial production.

On the other hand, the production of some protein polymers in a *K. phaffii* expression system is still challenging, due to the repetitive amino acid sequences that the polymers have in their structure, which can be target sites for proteases. It is known that proteolytic activity is minimized by growth on glycerol and glucose [98]. Therefore, some alternatives to the promoters widely used in expression of recombinant proteins in *K. phaffii* (such as P*AOX1*, induced by methanol or even constitutive promoters) have been explored. A copper-inducible promoter from *S. cerevisiae* (*CUP1*) was developed for gelatin production in *K. phaffii* [114]. This system offers the advantage of being expressed when cells are cultured in dextrose, an economical, non-toxic agent, and a non-flammable carbon source. This strategy provides an exploitable tool for the large-scale production of gelatin and other biomaterials in *K. phaffii*.

In view of this, the biomaterial production system in *K. phaffii* becomes promising for the most diverse biomedical applications, since several polymers of secreted proteins were successfully produced using this microorganism as a producer.

## 5. Applications in the Food and Feed Industry

One of the first uses of *K. phaffii* was in the food industry. Because of its natural ability to assimilate inexpensive methanol at high cell densities, *K. phaffii* was initially considered as an attractive food supplement in the form of single-cell protein (SCP). In this context, the development of a mutant strain with a high methionine content prompted British Petroleum Co to file a patent in the in the early 1980s [115]. The interest in SCP has recently re-emerged since methanol can be sustainably synthesized from CO_2_. Using adaptive laboratory evolution and metabolic engineering, Meng et al. [116] developed a *K. phaffii* strain with a protein content higher than other food sources such as soy, fish, meat, and whole milk.

A milestone in the use of food/feed products derived from *K. phaffii* was when the U.S. FDA awarded GRAS status to recombinant phospholipase C for degumming vegetable oils for food use in 2006 [10]. In 2016, Impossible Foods Inc. (Redwood City, CA, USA) launched the Impossible Burger, a plant-based alternative to traditional meat-based burgers. The Impossible Burger contains a recombinant protein produced in *K. phaffii* called leghemoglobin, a soy-derived heme protein similar to myoglobin. Heme proteins are important factors to mimic animal-derived meat flavors [117]. The Impossible Burger, a soy-derived product, has been proven safe for human consumption [118].

Today, the main interest in *K. phaffii* in the food and feed industry relies on the production of recombinant products, such as enzymes. The enzyme market is projected to reach USD 16.9 billion by 2027, growing at a compound annual growth rate (CAGR) of 6.8% from 2022 to 2027 [119]. This is due to the increasing use of enzymes as chemical substitutes, particularly in food and beverage, cleaning, and pharmaceutical applications. In the European Union, approximately 260 different enzymes are available, and most are produced by filamentous fungi (58%), yeast (5%), and bacteria (28%). A third of these enzymes are derived from genetically modified organisms. A list of the main enzymes produced in *K. phaffii* for the food and feed industry has been published elsewhere [119].

To enhance the digestibility of plant-based feedstuffs, the addition of phytases has been considered. Phytases reduce the need for inorganic phosphate supplements for monosgastric animals by removing phosphate from phytate, the main storage form of phosphorus in some plants. The production of recombinant phytase in *K. phaffii* is a good example of how this yeast-based platform can have a major impact in modern industrial biotechnology. According to a report by Validogen GmbH, the annual market for this enzyme is approximately USD 350 million [120]. In addition to phytase, xylanases are also a desirable supplement in feed since they reduce viscosity of raw plant material by degrading xylan. In order to reduce production costs, Roongsawang et al. [121] constructed an expression cassette formed by both a phytase and a xylanase coding gene separated by the 2A peptide sequence that promotes ribosome skipping. The results showed that the biochemical properties of the resulting enzymes were similar to those produced individually.

Although many commercial expression vectors are based on the strong inducible P*AOX1*, the presence of residual methanol in the final product is a matter of concern in the food industry. To avoid this, constitutive promoters or engineered P*AOX1* promoters may be used for enzyme production in a methanol-free medium [122]. Validogen GmbH has screened a promoter library of variants of the P*AOX1* and isolated a particular mutant that was able to secrete 20 g·L^−1^ phytase in non-methanol conditions as opposed to 22 g·L^−1^ in a methanol-induced control [120]. Bioprocess developments combining synthetic biology with metabolic engineering should contribute to further improving enzyme production in *K. phaffii*.

## 6. Advanced Tools for Synthetic Biology in *K. phaffii*

Synthetic biology is a relatively recent area that has made outstanding contributions to the bioeconomy worldwide, bringing innovative solutions to problems in diverse areas, and is now getting into the second decade of its life. Considering the several advantages of *K. phaffii* that place it as a desirable chassis organism for industrial applications [123], it is essential to discuss the main synthetic biology tools developed for this organism (Figure 1).

## 7. Synthetic Genetic Circuits

Synthetic genetic circuits aim to develop programmable organisms capable of performing a wide range of tasks [128]. Jacob and Monod first described endogenous genetic circuits, drawing parallels between electrical circuits and the gene expression control of lactose and tryptophan operons [129], with the first synthetic genetic circuit developed in 2000 [130,131]. These circuits are composed of modular genetic parts which need to be fully characterized, independent, reliable, orthogonal, tunable, composable, and scalable [132].

One of the first synthetic circuits reported in *K. phaffii* was a positive autoregulated circuit that aimed to obtain a methanol-free strain without gene deletion. The synthetic circuit consisted of the transcriptional activator Mxr1, constitutively expressed and acting in the derepression of P*AOX1*, and the Nrg1 repressor down-regulated by methanol. In another methylotrophic yeast, *Hansenula polymorpha*, the P*AOX1* orthologous P*MOX* is not glycerol sensitive and, as the difference between P*AOX1* and P*MOX* relies on its upstream transcriptional regulators, the authors hypothesized that up-regulation of the *MXR1* might lead to the same phenotype in *K. phaffii*. Therefore, they laced an extra *MXR1* copy under the control of P*AOX2*, a weaker promoter than P*AOX1*. As a result, using GFP expression assays, P*AOX1* started responding to the absence of glycerol without the need for methanol induction. Notably, to evaluate the viability of this circuit for recombinant protein production, a secreted single-chain variable fragment (scFv) was expressed under this system and showed a 98% increase in the presence of methanol and a 269% increase in the absence of glycerol [133].

A significant advance was the development of a malonyl-CoA-based regulated genetic circuit oscillator in *K. phaffii* [124]. This circuit consisted of two sensors based on the bacterial malonyl Coenzyme A (malonyl-CoA) system, in which malonyl-CoA binds to the repressor protein FapR and releases it to its DNA operator *fapO.* Sensor 1 comprises the FapR fused to the Prm1, a transcriptional activator of the P*AOX1.* In the presence of cerulenin, intracellular malonyl-CoA is upregulated, repressing the expression of the reporter gene. In the second sensor, the Prm1-FapR acts as a repressor instead of an activator. After validation, the authors designed the malonyl-CoA oscillator, allowing the conversion of the accumulated malonyl-CoA to polyketide. Because malonyl-CoA is the building block of several biochemical compounds, this oscillator might be the foundation stone for industrial and pharmaceutical applications.

Even with the promise of using synthetic genetic circuits as gene regulation tools, its design demands fulfilling several criteria, such as a synthetic genetic parts database for *K. phaffii* and landing pads to integrate them into biological systems, avoiding undesired endogenous interference [134]. Consequently, the availability of Genomic Safe Harbors (GSHs) [135], stable centromeric vectors [136], and synthetic chromosomes [137] become crucial.

## 8. CRISPR-Cas Systems as Tools for Gene Editing and Gene Regulation Control

In *K. phaffii*, the first CRISPR-Cas system evaluated was the CRISPR-Cas9 marker-less system, which paved the way for CRISPR-Cas9-based metabolic engineering in this organism [138]. Despite this, donor cassette integration via Homologous Recombination (HR) was still challenging. However, two years later, the same group demonstrated a CRISPR-Cas9 high-efficiency integration of marker-less donor cassettes via HR with the possibility of marker recycling in a Δ*KU70* strain [139], and the latter deletion was shown to increase HR efficiency [140]. These results expanded the CRISPR-Cas9 toolset for *K. phaffii*, allowing not only indels but also point mutations, deletions of genome sequence stretch, protein fusions, and the introduction of scarless tags, augmenting CRISPR-Cas9-based metabolic engineering possibilities significantly. As most industrial applications for *K. phaffii* require the integration of complex biosynthetic pathways, a multiloci genome integration tool is essential to further advancing this organism as a chassis biofactory host. Following this reasoning, Liu and collaborators showed a CRISPR-Cas9-based duplex and triplex integration in the Δ*ku70 K. phaffii* strain [125]. Nevertheless, a drawback of this tool was the low transformation efficiency with the increasing number of expression cassettes.

All the CRISPR-Cas9 systems mentioned above relied on ribozymes to express gRNAs, inserting a layer of complexity into their experimental design, and leading to the requirement to find *K. phaffii* RNA polymerase III promoters. These promoters were identified and successfully used in a multiplex genome editing strategy [141]. Expanding this system, CRISPR-ARE was developed for simultaneous gene activation, repression, and editing [142]. Recently, Go and collaborators developed another marker-less multiloci integration tool based on the CRISPR-Cas9 system [143]. They evaluated the integration efficiency of three genes of the β-carotene biosynthetic pathway in intergenic regions of a Δ*KU70 K. phaffii* strain. The results showed a slight increase in HR; however, the disruption of *KU70* impaired cell fitness and resulted in low transformation efficiency, indicating that the strategy was not the best option.

Another approach to increase HR in *K. phaffii* was the use of different exonucleases involved at the beginning of the HR process to Cas9 [144]. As proof of concept, the authors integrated genes of the fatty alcohol biosynthesis pathway with an HR improvement from approximately 66% to 91%. To overcome CRISPR-Cas9 requirement limitations, other CRISPR-Cas systems were evaluated. Zhang and collaborators developed a system based on Cas12a (Cpf1) that recognizes T-rich PAMs with a single CRISPR RNA (crRNA), shortening the gRNA expression cassette [145]. Despite this, the editing efficiency varied according to the target gene and diminished for triplex gene editing. Furthermore, its *off-target* potential was not evaluated. Recently, Liu et al. [126] described a highly programmable expression platform based on CRISPR-Cas systems (SynPic-X), and Deng and collaborators achieved the highest reported titer in *K. phaffii*, using CRISPR-Cas to regulate human lactalbumin (α-LA) production [146].

## 9. Conclusions

The diversity of compounds already produced in *K. phaffii*, as exemplified above, highlights the potential of this yeast to be employed as a microbial platform for the production of value-added chemicals, fuels, and bioproducts. The increasing knowledge in cell biology and physiology as well as the design of new synthetic biology tools for metabolic engineering certainly will support and contribute to the development of more robust strains and processes required for industrial application.

## Figures and Tables

**Figure 1 jof-10-00411-f001:**
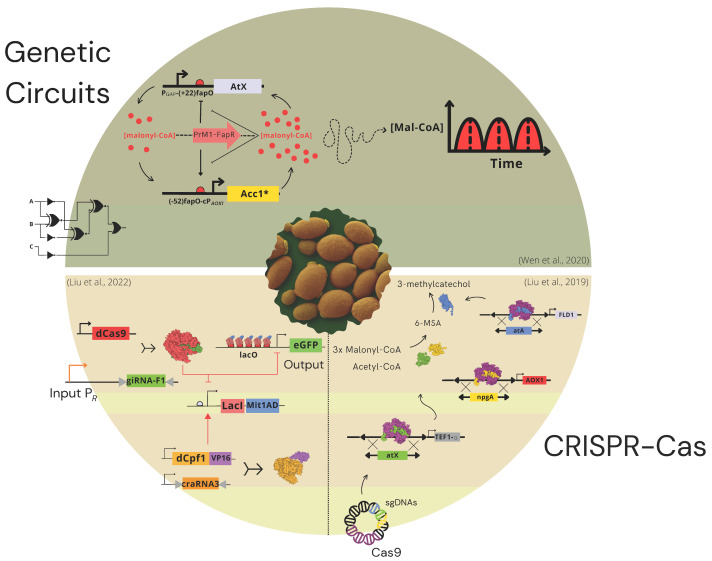
A schematic representation of the principal synthetic biology tools developed for the yeast *K. phaffii*, including synthetic genetic circuits and CRISPR-Cas systems. The genetic circuit topic is represented by the malonyl-CoA-based oscillator [124]. The CRISPR-Cas section includes the multiloci genomic integration tool [125] and the programmable expression platform SynPic-X [126]. The 3D design of the molecules was generated by Illustrate [127]. * In this genetic circuit Acc1 is a single base mutant which was shown to avoid deactivation by AMP-activated serine/threonine protein kinase (Snf1) upon glucose depletion in yeast.

**Table 1 jof-10-00411-t001:** Biopharmaceuticals produced in *K. phaffii* that are approved or undergoing clinical trials.

Brand Name (If Any)	Manufacturer/Research Institute	Biopharmaceutical	Commercial Approval/Clinical Trials	Reference
Kalbitor	Dyax Inc. (Burlington, MA, USA)	Ecallantide	2009 (USA)	[13]
Jetrea	ThromboGenics (Leuven, Belgium)	Ocriplasmin	2012 (USA) 2013 (Europe)	[14,15]
Semglee (USA and Europe) Insugen (India/Japan)	Mylan Pharmaceuticals (Morgantown, WV, USA) Viatris Ltd. (Galway, Ireland) Biocon (Bengaluru, India)	Biosimilar insulin glargine	2021 (USA) 2018 (Europe) 2016 (Japan) 2004 (India)	[16,17,18,19]
Inpremzia (Europe) Myxredlin (USA)	Baxter Holding (Utrecht, The Netherlands)	Biosimilar insulin	2022 (Europe) 2019 (USA)	[20,21]
Kirsty	Mylan Ireland Ltd. (Dublin, Ireland)	Biosimilar insulin aspart	2021 (Europe)	[22]
TTI-1612	Trillium Pharmaceutics (Brockville, ON, Canada)	Heparin-binding human epidermal growth factor (HB-EGF)	Phase I trials	[23]
Shanferon	Shanta Biotech (Hyderabad, India)	Interferon alpha 2b	2002 (India)	[24]
MFECP1	University College Medical School (London, UK)	Anti-carcinoembryonic antigen antibody–carboxypeptidase G2	Phase I trials	[25]
Vyepti	Lundbeck Pharmaceuticals (Copenhagen, Denmark)	Eptinezumab	2020 (USA) 2022 (Europe)	[26]
Clazakizumab	Bristol-Myers Squibb (New York, NY, USA)	Clazakizumab	Phase II trials	[27]
Shanvac-B	Shanta Biotech	Hepatitis B vaccine	1997 (India)	[28]
Recombinant Na-GST-1	Baylor College of Medicine (Houston, TX, USA)	Hookworm vaccine	Phase I trials	[29]
Sm-TSP-2	National Institute of Allergy and Infectious Diseases (Bethesda, MD, USA)	Intestinal schistosomiasis vaccine	Phase I trials	[30]
PfAMA1-DiCo	Institut National de la Santé et de la Recherche Médicale (Paris, France)	Malaria vaccine	Phase I trials	[31]

**Table 2 jof-10-00411-t002:** Representative bio-based compounds produced by *K. phaffii* from different substrates.

Product	Substrate	Main genetic Modifications and Strategies	Process	Application	Production (g·L^−1^)	Yield (g·g^−1^)	Reference
** *Organic acids* **							
Lactic acid	Glycerol	Integration of heterologous lactate dehydrogenase (LDH) and overexpression of endogenous lactate transporter	Fed-batch	Food, pharmaceutical, textile, and chemical industries	~28	0.7	[73]
	Methanol	Multicopy integration of LDH	Batch		3.48	0.22	[80]
Malic acid	Glucose and methanol	Combined expression of *pc* and *mdh1*	Batch	Food, pharmaceutical, and chemical industries	42.28	-	[81]
3-hydroxypropionic acid	Glycerol	Expression of an engineered *mcr*; overexpression of *ACC1*, *ACS*, *ALD6*, *PDC1*; deletion of *ArDH*	Fed-batch	Building block for production of chemicals, such as acrylic acid and biopolymers	24.75 to 37.05	0.13 to 0.194	[82,83]
Xylonic acid	Xylose	Expression of xylose dehydrogenase (XDH)	Batch	Building block, cleaner agent, and cement additive	37	0.96	[70]
** *Sugar alcohols* **							
Inositol	Glycerol and glucose	Heterologous protein expression and regulation of carbon flux of glycolysis and pentose phosphate pathway	Fed-batch, high cell density	Pharmaceutical, food and feed industry	30.71	-	[84]
Xylitol	Xylose	Expression of xylose reductase (*XYL1*) from *Scheffersomyces stipitis* and *gdh* from *Bacillus subtilis*	Biotransformation	Sweetener with application in food, oral and personal care industries	320 (mM)	80%	[74]
** *Biofuels and oleochemicals* **							
2,3-butanediol	Glucose	Expression of *alsS* and *alsD* genes from *Bacillus subtilis*	Fed-batch	Chemical platform, biofuel	74.5	0.3	[85]
Isobutanol	Glucose	Expression of *LlkivD*, *ScADH7*, *PpIlv2*, *PpIlv3*, *PpIlv5*, *PpIlv6*	Batch	Chemical platform, biofuel	2.22	0.22	[78]
Free fatty acids (FFA)	Methanol	Deletion of fatty acyl-CoA synthetase genes; expression of Mm*ACL*, *BbPK*—Ck*PTA*; overexpression of *DAS2*	Fed-batch	Platform for oleochemical and biofuel production	23.4	0.078	[86]
** *Terpenoids* **							
Dammarenediol-II	Methanol and glycerol	Expression of Pg*DDS*; overexpressing *ERG1*; down-regulation of *ERG7*	Batch	Bioactive compound for pharmaceutical industry	-	0.736 (mg.gDCW^−1^)	[87]
Lycopene	Glucose	Expression of heterologous carotenogenic enzymes and regulation of lipid metabolism pathway	Batch	Nutraceutical supplements, food colorants, cosmetic and pharmaceutical industries	7.24	75.48 (mg.gDCW^−1^)	[88]
β-carotene	Glucose	Expression of heterologous carotenogenic enzymes crtE, crtB, crtI and crtL	Batch		-	339 (μg.gDCW^−1^)	[76]
(+)-nootkatone	Glucose and methanol	Co-expression of heterologous genes *HmHPO*, *AtCPR*, and *CnValS*;	Fed-batch	Flavor and fragrance compound for the food and cosmetics industries	0.2		[89]
		Overexpression of endogenous Adh and truncated Hmg1					
** *Polyketides* **							
6-Methylsalicylic acid	Methanol	Overexpression of *atX* and *npgA*	Fed-batch	Bioactive compounds for pharmaceutical industry	2.2	-	[90]
Lovastatin and Monacolin J	Methanol	Expression of *LovB*, *LovC*, *LovG*, *NpgA*, *LovA*, *CPR*, *LovD*, *LovF*	Fed-batch; co-culture		0.250 and 0.593	-	[75]
** *Biopolymers* **							
Polyhydroxyalkanoates	Oleic acid	Expression and peroxisomal targeting of *Pa*PHA synthase	Batch	Biobased and biodegradable thermoplastic polyesters with broad-range industrial application	1%.gDCW^−1^	-	[91]
Hyaluronic acid	Glucose and methanol induction	Overexpression of *xhasA2*, *xhasB* from *Xenopus laevis* and endogenous *hasC*, *hasD*, *hasE*	Fed-batch	Biopolymer used in medical and pharmaceutical industry	0.8–1.7	-	[77]
Chondroitin sulfate	Methanol	Expression of *kfoA*, *kfoC*, *tuaD*, C4ST, ATPS and APSK	Fed-batch	Medical and nutraceutical industry	2.1	-	[65]
Heparin	Methanol	Expression of *tuaD*, *kfiC*, *kfiA*, NDST, C5 epi, 2OST,3OST, 6OST	Fed-batch	Anticoagulant drug	2.08	-	[92]

## Data Availability

Not applicable.

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
