# Peer review of "Applications of the Methylotrophic Yeast *Komagataella phaffii* in the Context of Modern Biotechnology"

_jof, 2024, doi:10.3390/jof10060411_

Round 1

Reviewer 1 Report

The manuscript (MS) is probably compiled in a hurry. It is bulky and largely descriptive. I recommend to  shorten the MS by about 30% by taking out the concerning characteristics making K. phaffii attractive host system for industrial applications, its physiological features, vectors, promoters, advantages over other yeasts and bacteria and limitations should not be addressed. This is mostly written in Introduction part (lines 29-90). Therefore, new Introduction (and Abstract) should be written. The paragraphs ’Advanced tools for synthetic biology in K. phaffii’, ’Synthetic genetic circuits’ and ’CRISPR-Cas systems as tools for gene editing, and gene regulation control’ should be condensed. Currently this text is a bit long-winded.

1) In many cases, cited references were erroneous (see for example the lines 42, 64, 76). Therefore I suggest that the authors should carefully read the entire text and verify all citations.  

2) List of references should be revised and the entries unified for style.

3) The MS contains numerous passages with confusing or inaccurate text. See for example lines 69-71, 81-82 (constitutive genes and inducers?), 88-89, 91-97, 110-111, 484 (methanol-free strain?) etc. I recommend to conduct thorough revision of the text to improve the language and readability.

4) Title of the Table 1: I recommend to replace  ’Biologicals’ with ’Biopharmaceuticals’.

5) FFA in Table 2 (left column) should be written in full.

6) The text concerning the data presented in Figure 1, should refer to this figure.

7) Sectors of Figure 1 show references such as Liu et al., 2019, Liu et al., 2022 etc. As this MS has numerical citations, respective papers are not easily found. The authors should solve this problem.

8) In the first part of the MS, the citations are in square brackets, later on not. Style should be unified.

9) The MS has many authors, but their contribution (line 581) is not given.

Reviewer 2 Report

Line 74: I would recommend checking this sentence again as another paper suggests different: Vogl et al., A Toolbox of Diverse Promoters Related to Methanol Utilization:

Functionally Verified Parts for Heterologous Pathway Expression in

Pichia pastoris, 2015

Line 101: This is not correct: This yeast received GRAS status by the American Food and Drug Administration in 2006 [29]. Looking at reference 29 it is clearly stated that the product, Phospholipase C, is generally recognized as safe and not the organism. I think in general it is not possible that an organism gets the GRAS status. 

Line 446: The sentence needs rephrasing. To the best of my knowledge 2A sequences result in ribosomal skipping. Thus there is no cleavage as the phytase and xylanase are never connected on a protein level.

Line 467: the resolution of Figure 1 should be improved.

Line 529: I think it should be briefly explained what the ku70 deletion facilitates and a reference should be provided.

Line 16: Pichia pastoris  Pichia pastoris

Line 65 and the following: I know one can find this in many recent papers but to be correct one expresses a gene, and one produces a protein. I think we should start to be more careful about this. 

Line 161: Still  Still, 

Line 185: The . is shown in red.

Line 219 and following: CO2  CO2

Line 227: Typically, MUT refers to methanol utilization, not methanol assimilation.

Line 243: Komagataella phaffii  K. phaffii

Line 273 and 395: I suggest rephrasing. If the authors write “on the other hand” there has to be “on the one hand” before in the text. 

Line 401: Saccharomyces cerevisiae  S. cerevisiae

Line 445: et al  et al.

Line 454: The authors should maintain the same style throughout the manuscript: g/L vs. g l-1 vs. g.l-1

Line 458: K. phaffii K. phaffii 

Line 485: ofa  of a

Line 491: result,using  result, using

Line 563: and elsewhere: et al. vs. et al. should be identical throughout the manuscript.

Round 2

Reviewer 1 Report

The authors have revised the manuscript and it looks much better now. 

Table 2. Lactic acid (not Lactic Acid).

Lines 496-498 (Data availability statement). Why conflicts of interests here? I consider that it would be appropriate to write here just 'not applicable' as it is a review article.

Author Response

We have adjusted the MS following the minor corrections suggested by reviewer #1 to whom we would like to express our gratitude for his comments which enriched our MS so much